# Natural and Synthetic Biomaterials for Engineering Multicellular Tumor Spheroids

**DOI:** 10.3390/polym12112506

**Published:** 2020-10-28

**Authors:** Advika Kamatar, Gokhan Gunay, Handan Acar

**Affiliations:** 1Stephenson School of Biomedical Engineering, University of Oklahoma, Norman, OK 73019, USA; Advika.V.Kamatar-1@ou.edu (A.K.); Gokhan.Gunay-1@ou.edu (G.G.); 2Stephenson Cancer Center, University of Oklahoma, Norman, OK 73104, USA

**Keywords:** MCTS, multicellular tumor spheroid, three-dimensional culture, biomaterials, microenvironment, hydrogel, scaffold, biopolymers

## Abstract

The lack of in vitro models that represent the native tumor microenvironment is a significant challenge for cancer research. Two-dimensional (2D) monolayer culture has long been the standard for in vitro cell-based studies. However, differences between 2D culture and the in vivo environment have led to poor translation of cancer research from in vitro to in vivo models, slowing the progress of the field. Recent advances in three-dimensional (3D) culture have improved the ability of in vitro culture to replicate in vivo conditions. Although 3D cultures still cannot achieve the complexity of the in vivo environment, they can still better replicate the cell–cell and cell–matrix interactions of solid tumors. Multicellular tumor spheroids (MCTS) are three-dimensional (3D) clusters of cells with tumor-like features such as oxygen gradients and drug resistance, and represent an important translational tool for cancer research. Accordingly, natural and synthetic polymers, including collagen, hyaluronic acid, Matrigel^®^, polyethylene glycol (PEG), alginate and chitosan, have been used to form and study MCTS for improved clinical translatability. This review evaluates the current state of biomaterial-based MCTS formation, including advantages and disadvantages of the different biomaterials and their recent applications to the field of cancer research, with a focus on the past five years.

## 1. Introduction

Lack of translation from the results of preclinical research to clinical trials has been a major problem in cancer research. Two-dimensional (2D) cell culture and mouse models have long been the standard for studies of cancer biology and drug screening and have greatly improved our understanding of cancer’s complexities. However, 2D culture exposes cells to a rigid plastic surface on one side and to a liquid on the other as cells grow in a flat monolayer, altering the cytoskeleton and causing abnormality in cellular functions such as metabolism and protein expression [1]. Mouse models are more relevant than 2D culture to complex human biology and have greatly enhanced our understanding of tumor biology, but they suffer from high cost and resource requirements [2]. Ethical review boards encourage researchers to at least reduce or refine, if not entirely replace, their animal usage in order to harm as few animals as possible—a principle known as “the three Rs” [3]. Furthermore, mouse models do not completely reproduce the complexity of human physiology and metabolism, which may contribute to the fact that less than 8% of medicinal compounds entering Phase I trials eventually make it to market as noted in a 2004 Food and Drug Administration (FDA) report [4,5]. Alternatives to 2D culture and mouse models include clinical samples, but their use is limited by intratumoral heterogeneity and are governed by many federal regulations [6], making them low-throughput and difficult to routinely use.

Recent advances in 3D cell culture, such as cell culture in hydrogels that mimic the extracellular matrix, have attempted to bridge the gap between preclinical and clinical results. In particular, multicellular tumor spheroids (MCTS) have emerged as a promising in vitro model for cancer research because of their high throughput, low cost and increased physiological relevance compared to 2D culture. MCTS are defined as 3D spherical clusters of malignant cells [7], and have been shown to mimic many features of solid tumors in vivo, such as cell–cell and cell–extracellular matrix (ECM) interactions [8,9], increased drug resistance [10,11,12], cell polarity [13], and nutrient diffusion gradients [14].

Studies on MCTS began in the 1970s and 1980s, led by the efforts of R.M. Sutherland and colleagues [15,16]. In metastatic tumors, lack of vascularization leads to areas of necrosis of cells with nutrient deficiency, hypoxia, and acidity [16,17,18]. Tumors also exhibit layers of proliferating, quiescent (non-proliferating), and necrotic cells at progressively greater distances from blood vessels [16]. As we demonstrated in Figure 1, it is now well established that MCTS replicate these steep nutrient, oxygen and pH gradients and cell layers in those typically above 500 μm in diameter (Figure 1) [19]. Another hallmark of metastatic cancer is the epithelial-to-mesenchymal transition (EMT), in which epithelial cancer cells lose cell–cell adhesions such as E-cadherin, apical-basolateral polarity, and take on a mesenchymal phenotype to promote migration and invasion [13]. MCTS culture, as opposed to 2D culture, can induce this EMT [20] and affect the migratory profiles of MCTS [12], which is crucial in studies attempting to understand and prevent metastasis. In fact, in ovarian cancer, metastasis is proposed to occur through detachment of cells into the peritoneal cavity, where they aggregate into MCTS and create malignant ascites [13], further highlighting the direct relevance of the MCTS model to human cancer pathophysiology.

There are two broad strategies for preparation of MCTS as an in vitro model for drug screening and cancer biology investigation: matrix-free formation and matrix-dependent formation. Matrix-free MCTS formation techniques include culture via liquid overlay, hanging drop, rotating spinner flasks, and magnetic levitation [21,22]. Matrix-free techniques have been widely used due to their high throughput, high reproducibility, and relative ease of culture, but they do not include tumor extracellular matrix (ECM) during initial assembly, which provides important physicochemical cues in tumor development in vivo [23,24]. Matrix-dependent techniques, in contrast, involve the use of biomaterials to recreate these in vivo cell–ECM interactions. It should be noted that microfluidics has also been considered a promising strategy for MCTS formation and culture, and this technique often incorporates polymers that are discussed throughout this review. Engineering considerations for microfluidic MCTS formation have been extensively reviewed elsewhere [25], and will not be a focus here. The purpose of this review is to provide a practical guide for choosing a technique for MCTS formation in vitro. Matrix-free and matrix-dependent techniques for MCTS formation are explained with an emphasis on biopolymer matrix-dependent techniques. Advantages and disadvantages of each biopolymer and their recent applications are explained.

## 2. Matrix-Free Multicellular Tumor Spheroids (MCTS) Formation

Matrix-free MCTS formation occurs through self-aggregation of cells. During this process, cells also produce certain components of ECM and organize their own 3D structure. This provides a good platform to study completely endogenous ECM. Matrix-free techniques include liquid overlay, hanging drop, spinner flask and magnetic levitation, as reviewed briefly below. Each technique is depicted in Figure 2, and a comparison of the cost, formation timeline and uniformity of both matrix-free and matrix-dependent techniques can be found in Table 1. Values for diameters ± standard deviations represent the reported size on the last day of culture of MCTS. However, readers are advised to evaluate these results based on their particular application because uniformity and day of formation vary based on the cell line, polymer concentration, and even the initial seeding density; for example, Nakod et al. reported that increasing the initial seeding density influenced the size distribution of MCTS in the glioblastoma multiforme (GBM) [26]. In addition, to the best of the authors’ knowledge there is no definition of when an aggregate becomes an MCTS, so many papers do not report the exact day that MCTS “formed,” but rather monitor their culture over several days as they increase in compactness from initial seeding to a spherical body.

### 2.1. Liquid Overlay Technique

The liquid overlay technique (LOT), sometimes referred to as the ultra-low attachment (ULA) technique, is based on the use of non-adhesive plastic surfaces or functionalizing the surfaces of tissue culture well plates with non-adhesives (hydrophilic polymers) [21]. For example, poly (2-hydroxyethyl methacrylate) (HEMA) treated surfaces resulted in hepatocyte spheroid formation [41], and ATDC5 chondrocyte cells formed spheroids on polyethylene glycol (PEG) coated surfaces [42]. Elimination of cell–surface interactions allows increased cell-to-cell interaction and initiates MCTS formation (Figure 2A). In addition to its low cost and relatively easy handling, one major advantage to using the LOT is that it provides post-processing directly in the plates where MCTS are formed [9,43], making it very useful in high-throughput experiments [44]. Typical MCTS formation with this technique involves functionalizing the surface with either biomaterials or commercially available anti-adhesive solutions, followed by washing the surfaces and seeding cells. MCTS formation varies among cell types and should be optimized for different cell lines. Exchanging 50% of the culture media offers fresh medium for the MCTS while preventing the loss of MCTS due to accidental removal of the MCTS along with the medium.

LOT frequently uses microwells as a culture platform because they offer a defined structure for consistent, reproducible, and size-controllable MCTS formation. Several commercial products are available to generate MCTS in microwells via LOT, including Aggrewell™ plates (Stemcell™ Technologies) and Elplasia^®^ plates (Corning^®^). These products free the researcher from tedious traditional microwell fabrication but lack the flexibility to change well structure or volume depending on the application. In response to these shortcomings, researchers have fabricated their own microwells using non-adhesive biomaterials with the goals of geometrical symmetry, sufficient well volume, low surface roughness, and low cost [45]. For example, microwell formation via CO_2_ laser ablation of polystyrene has been shown to produce size-controlled A549 lung cancer MCTS [45]. In addition to polystyrene, agarose is a very common non-adhesive biomaterial used to form microwells, and allows MCTS formation without any additional surface modification [46,47,48].

LOT is one of the most common matrix-free approaches and has enabled important progress in studies on cancer stem cell formation and chemoresistance. For example, the LOT was recently used to study the effect of size and shape of ovarian carcinoma MCTS on drug resistance to Cisplatin and Taxol, as well as migration capacity of different MCTS morphologies [12]. However, not all cell lines are able to form MCTS via this technique, even within the same type of cancer. For example, while OVCAR8 and OVCAR3 cell lines formed compact and loose spheroids, respectively [12], the ovarian carcinoma line SKOV-3 did not form MCTS similar to those found in patient ascites, possibly due to the different cell surface receptors expressed on different cell lines [49]. Thus, LOT requires a well-researched choice in cell lines. Because of the ease of sample preparation and low cost, as well as having samples with ECM structures relevant to their physiological counterparts, LOT is one of the most reliable and commonly used technique for the cell lines that are known to form aggregates.

### 2.2. Hanging Drop Technique

The hanging drop technique requires no special equipment and relies on preparation of desired numbers of cells/drop and provides reproducible MCTS formation [50]. Cell suspension drops (15–30 mL) are placed onto the underside of the lid of a tissue culture plate (Figure 2B). When inverted, surface tension keeps the drop intact and forces cells to localize at the bottom of the drop [21]. One disadvantage of this technique is that the MCTS have to be transferred prior to post processing [51]. Additionally, volumes used to culture MCTS are very small volumes (10–15 mL), so evaporation is a major problem [52]. A comparison between LOT and the hanging drop technique illustrated the ease of handling and monitoring when using LOT [27].

### 2.3. Spinner Flask Technique

The spinner flask technique allows cell to cell interaction by preventing the settling of the cells through continuous spinning [53], and leads to MCTS formation (Figure 2C). For example, HepG2 hepatocellular carcinoma cell suspensions were seeded at 1 × 10^5^ cells/mL into 500 mL siliconized spinner flasks and stirred at 60 rpm for 4–6 weeks. The resulting MCTS were 200–300 μm diameter and were transferred into agarose-coated well plates for further processing [54]. In another study, rat liver hepatocytes were cultured at 5 × 10^5^ cells/mL in a 250 mL spinner flask for 6 h, resulting in 80% aggregate formation [55]. The speed of the continuously rotating flask is important to consider; low speeds might result in settling of the cells and high speeds might harm the cells due to shear stress. One advantage of using the spinner flask is the ability to produce large quantities of MCTS. However, during continuous spinning MCTS formation cannot be visualized [53], which is important in monitoring morphology and size.

### 2.4. Magnetic Levitation Technique

The magnetic levitation technique enables cells to form MCTS through applying magnetic field [22]. Prior to magnetic force application, cells of interest are loaded with magnetic nanoparticles and the application of magnetic force to draw cells together facilitates cell–cell adhesion and, subsequently, MCTS formation (Figure 2D). In addition to forming tumor models [56], it has also been used for adipose tissue [57] and bronchiole culture [58]. The magnetic levitation technique, in addition to MCTS formation, also offers MCTS fusion; astrocyte MCTS and glioblastoma MCTS were fused within 12 h by using this technique [22]. Magnetic levitation offers a platform where MCTS are formed in a short period of time relative to other approaches (e.g., 24 h for magnetic levitation compared to 7 days in Matrigel culture [56]), which allows faster imaging [59]. It must be considered that, even though magnetic nanoparticles have been shown not to affect cell behavior or an inflammatory response within 30–500 G magnetic field [22,58], a higher magnetic field (800–400 G) has been shown to affect cell behavior [60].

## 3. Matrix-Dependent MCTS Formation

While matrix-free techniques are frequently used due to their reduced labor requirements and high throughput, scaffold-dependent MCTS formation presents significant advantages in mimicking the tumor ECM, facilitating more physiologically relevant conclusions. The ECM plays a prominent role in cell behavior and regulation in vivo. Dynamic reciprocity describes the continuous, bidirectional interactions between cells and their surrounding matrix [61], in which cells secrete enzymes such as matrix metalloproteinases (MMPs) that remodel nearby polymers [62]; in turn, these polymers exert mechanical forces on the cells, producing changes in biochemical signals and thus gene expression through mechanotransduction [63]. Matrix-dependent techniques include various natural and synthetic polymers during MCTS formation, whether as a hydrogel, solid scaffold or microbead, to better mimic these cell–ECM interactions and the nutrient, pH, and oxygen gradients observed in solid tumors in vivo.

Deciding whether to use a matrix-dependent technique depends on the parameters of interest, the cell lines used, and the downstream uses for the MCTS. For instance, studies of ECM remodeling, invasion or angiogenesis will require a matrix, either through MCTS formation directly within a scaffold or through matrix-independent formation and subsequent embedding into a scaffold. Furthermore, some cell lines do not form MCTS via matrix-free techniques, but use of ECM can help cell aggregation [64]. The cell lines that produced MCTS with matrix-dependent techniques are presented in Table 2.

Matrix-free MCTS are also easily disrupted during culture; for example, when changing media on MCTS cultured using LOT in Aggrewell™ microwell plates, precise pipetting angle and speed is necessary to avoid disruption of the well and loss of the MCTS. However, use of a scaffold frequently requires subsequent recovery of MCTS for further experimentation, such as cell counting, immunostaining, flow cytometry and Western blotting [65]. This is typically carried out using a cell recovery solution that separates MCTS from the scaffold without compromising their 3D architecture and adds additional complication and risk to the MCTS compared to a matrix-free approach.

The goal of matrix-dependent MCTS culture is to mimic the native ECM as closely as possible while maintaining precise control over experimental parameters such as cell number, cluster shape, mechanical stiffness, and porosity [66]. To control these factors, a variety of scaffold modification strategies have been developed. For example, an important parameter in engineering the tumor ECM is cell adhesion. In contrast to natural polymers, polysaccharide and synthetic polymers (e.g., hyaluronic acid, alginate, chitosan, PEG and poly(lactic-co-glycolic acid) (PLGA)) require additional modifications with integrin-binding peptide domains, such as arginine-glycine-aspartic acid (RGD) [67], isoleucine-lysine-valine-alanine-valine (IKVAV) [68] and tyrosine-isoleucine-glycine-serine-arginine (YIGSR) [69] in order to induce MCTS formation [68]. Composite approaches, in which two or more polymers are used together as a matrix, are also common to achieve greater control over parameters such as stiffness. For example, in a composite Matrigel^®^-PEG hydrogel, the crosslinking density of PEG was varied to increase the gel rigidity, which cannot be accomplished with Matrigel^®^ alone; furthermore, α-cyclodextrin-conjugated RGD was incorporated to study dissemination and adhesion of normal and malignant mammary epithelial cells [70].

Natural polymers, such as collagen and collagen-derivatives (e.g., gelatin), hyaluronic acid, and Matrigel^®^, have been used most widely because of their inherent cytocompatibility and ability to be remodeled by cells [71]. In contrast, synthetic polymers, such as PEG, allow more precise control over experimental parameters such as hydrogel stiffness and ligand density. The following sections are intended to provide a brief overview of the biopolymers commonly used in MCTS formation, their recent uses, and advantages and disadvantages to consider. Also, studies conducted on different cancer cell lines using the matrix-dependent techniques discussed in this section, with a concentration on studies conducted in the past five years, are given in Table 2.

### 3.1. Natural Biopolymer Matrices

Common natural polymers in 3D culture of MCTS include collagen, hyaluronic acid, Matrigel^®^, alginate and chitosan. Among these, collagen and Matrigel^®^ contain natural epitopes that allow cell–matrix interactions similar to those found in vivo. In contrast, alginate/chitosan scaffolds are bioinert and can be functionalized to facilitate interaction with cell-surface receptors. These variable cell–matrix interactions, along with parameters such as mechanical stiffness and tunability, determine the appropriate natural biopolymer matrix to use for MCTS formation.

#### 3.1.1. Collagen-Based Matrices

Collagen is the most abundant protein of the human ECM [86]. It is formed from a proline-rich polypeptide triple helix and is degraded by a group of matrix metalloproteinases (MMPs) known as collagenases [62]. Collagen facilitates cell adhesion through its integrin-binding domains such as glycine-phenyalanine-hydroxyproline-glycine-glutamic acid-arginine (GFOGER) peptide, and collagen derivatives such as gelatin are recognized by different sequences such as RGD peptide [87]. Because of its biological relevance and its straightforward isolation [88], collagen is commonly used in biomedical applications ranging from tissue engineering to cancer research.

Because the tumor extracellular environment is composed mainly of collagen I fibers [89], collagen hydrogels are common scaffolds for MCTS study. For example, collagen hydrogels have been used as drug-screening platforms [90], as an environment to study invasion [91], and to induce MCTS formation [91,92]. However, a more frequently used approach is to embed MCTS formed via a matrix-free technique into collagen gels for subsequent studies, such as migration and invasion [90,93,94]. Most recently, collagen is gaining use in MCTS formation via a microfluidic approach, which has been successful across many cancer types, including colorectal [80], breast [34], and lung [84].

In addition, collagen scaffolds are particularly useful in studies of the cancer stem cell population within tumor models. Cancer stem cells are a fraction of cells from a tumor that possess self-renewal and differentiation capabilities that allow them to generate new cancer structures [95]. Rat tail type I collagen coated on well-plates at 40–60 μg/mL has been shown to convert cancer cells into cancer stem cells [96], and when used as a scaffold for MCTS type I collagen has enhanced the stem cell quantities of the MCTS, increasing their similarity to a tumor in vivo. For example, a recent study formed MCTS from MDA-MB-231 and MCF7 breast cancer cells via LOT and embedded them into a collagen gel formed using 4 mg/mL of high concentration rat tail type I collagen (BD Biosciences). They observed an increased cancer stem cell population in the MCTS core compared to the MCTS periphery, diffused cells within a collagen gel, and cells in 2D culture, which likely corresponds to the observed higher drug resistance of the MCTS core towards paclitaxel and cisplatin despite equal drug penetration [97]. Another study cultured U87 and primary human glioma cells in scaffolds formed using bovine collagen membranes isolated from spongy bone and observed increased sphere formation and stemness markers in cells cultured in 3D compared to 2D [72].

Collagen usually shows low immunogenicity and is well tolerated and biodegradable in spite of its animal source, but batch-to-batch variability and the dependence of the gel microarchitecture on collagen concentration, temperature, pH, and ion concentration can make reproducibility and strict control of experimental parameters difficult [98]. Because of its natural source from ECM, collagen is one of the few polymers that may induce stemness properties in cancer cells similar to the stemness behavior observed in vivo that contributes to tumor heterogeneity [72]. Thus, its greatest strengths as a biopolymer for MCTS formation lie in its biological relevance, natural adhesion sites, and its popularity, which has led to a large library of studies that serve as a source of information on MCTS formation in various cell lines using collagen.

#### 3.1.2. Hyaluronic Acid-Based Matrices

Hyaluronic acid (HA, also known as hyaluronan), is a non-sulfated glycosaminoglycan with a high molecular mass (between 1–10 MDa) and a length of 2–20 μm in native tissue [99]. HA is a critical and abundant ECM component, and can thus activate specific cell signaling receptors that influence cell behavior. For example, HA interacts with the transmembrane receptor CD44, which promotes cell adhesion and migration [100]. The role of HA and its two primary cell surface receptors, CD44 and receptor for HA-mediated motility (RHAMM) in inflammation and tumorigenesis is complex and is under active investigation as a therapeutic strategy [99]. However, natural HA is hydrophilic and thus non-adherent, which encourages cell–cell adhesion [101] and MCTS formation.

HA is the main component of native brain ECM and plays a significant role in signaling pathways and tumor progression [102]. Because of its biological relevance and its ability to mimic the stiffness of the native brain environment, which is between 200–1000 Pa [103], HA-based hydrogels have been used to study MCTS culture of GBM. A recent study found that culturing the U87 cell line and patient-derived D456 GSCs (glioblastoma stem cells) in HA-methacrylate or RGD-modified HA-methacrylate hydrogels resulted in MCTS formation and increased expression of stemness markers compared to monolayer and suspension culture in both serum-free and serum-containing media [26]. In fact, HA incorporation can induce GBM MCTS formation in 3D environments that otherwise do not permit cluster growth, such as poly (ethylene glycol)-tetraacrylate (PEG4A) [104]. In addition to use as a matrix for MCTS formation, HA-based hydrogels have been used to study the behavior of GBM MCTS cultured using matrix-independent approaches [105,106]. In these studies, GBM MCTS were formed using the hanging-drop method before seeding into hydrogels containing HA for further study on migration [105] and morphology [106]. In the latter study by Ananthanarayanan et al., a tumor MCTS invasion assay was conducted using an RGD-functionalized HA-methacrylate hydrogel to compare the migration profiles of U87-MG and U-373-MG human glioblastoma cell lines. Importantly, they observed different modes of motility in the HA gels compared to their previous studies in 2D or collagen-based gels, which were similar to the motility behavior observed in brain slice cultures, indicating the importance of a biologically relevant 3D in vitro environment to mimic in vivo tumor behavior [106].

HA has also been used to study numerous other tumor types, including prostate [107,108], breast [68,76,109], colon [109], and ovarian [109]. For example, Tang et al. tuned the stiffness of HA-alginate composite hydrogels by varying the volume ratio of HA, and seeded PC-3 and DU145 prostate cancer cells into the solution. They observed MCTS formation in both cell lines, and upon testing of seven anticancer therapeutics between 2D, 3D and an in vivo murine model they found that the 3D HA-alginate tumor model much more closely represented the drug efficiency in the in vivo model [107]. In a recent study, Antunes et al. formed heterotypic MCTS with PC-3 prostate cancer cells and human osteoblasts (hOBs) to mimic prostate cancer metastasis to bone by using covalently crosslinked methacrylated hyaluronic acid (HA-MA) and gelatin-methacryloyl (GelMA) assembled into spherical microgels on superhydrophobic surfaces [24]. In HA-only gels they observed generalized cell death, likely due to lack of cell adhesion peptides (RGD, IKVAV or YIGSR), but incorporation of GelMA into the microgel increased metabolic activity to more closely mimic the proliferative activity of prostate cancer metastasis to bone [110]. In drug-screening studies with cisplatin, the heterotypic PC3-hOB spheroids showed a greater drug resistance in 3D microgels than scaffold-free MCTS. Baker et al. incorporated IKVAV domains at 35 nmol IKVAV/mg HA into a 1.1% HA-based scaffold functionalized with methyl furan and aldehyde motifs and observed optimal breast cancer MCTS formation, although the same effects were not observed with RGD or YIGSR [68]. These results indicate the importance of (1) considering cell adhesion motifs when choosing a scaffold; and (2) ECM inclusion during drug screening in order to obtain a more accurate assessment of treatment performance.

While many studies have chosen to use HA in studies of brain tumors because of its abundance in normal brain tissue, it is important to note that the glioma ECM differs greatly from the normal brain and contains large quantities of fibrillary collagens [72]. Therefore, it has been proposed that collagen may serve as a more physiologically relevant biomaterial for glioma studies [72]. However, HA is cheaper than collagen [111], and biotechnological approaches have enabled production of large quantities of high-purity HA via bacterial fermentation [112], avoiding much of the immunogenic risk associated with collagen [98]. Many studies have shown that increased HA deposition is correlated to unfavorable outcomes in cancer types such as colon [113], breast [114], lung [115], and stomach [116], indicating that it is an important polymer to consider when designing a scaffold for tumor modeling. In addition, in diseases such as cancer, the native high molecular weight HA (HMW HA) can be degraded into lower-weight fragments (LMW HA) and HA with different molecular weights can affect cancer cells differently [101]. However, the effect of the size of HA molecules is not yet well understood, which may add an extra element of variability during experimental design and merits further study. Finally, it should be explicitly mentioned that HA is soluble at room temperature and undergoes rapid turnover in the body [117], and thus requires chemical modifications in order to crosslink this polymer into a hydrogel for 3D culture [24]. The options and strategies for HA chemical modification have been explained in depth elsewhere [117].

#### 3.1.3. Matrigel^®^-Based Matrices

Basement membrane is a specialized form of ECM that separates tissue epithelium from the stroma. It provides structural support and compartmentalizes the given tissue and is an important regulator of cell behavior and tumor angiogenesis [118]. Matrigel^®^ is a commercialized product consisting of basement membrane extract from Engelbreth–Holm–Swarm (EHS) murine tumors, and has been used in studies of cancer biology for over four decades [119]. It consists of many ECM components, but primarily laminin, collagen IV, entactin, and heparan sulfate proteoglycan [120] as well as several growth factors such as fibroblast growth factor, epidermal growth factor, insulin-like growth factor 1, transforming growth factor beta, platelet-derived growth factor and nerve growth factor [121] (a reduced growth factor version of Matrigel^®^ is now commercially available) [122]. Matrigel^®^ has been used for many applications, from MCTS formation [65] to injection with patient-derived xenografts to improve take and growth [123,124].

Along with collagen and decellularized ECM, Matrigel^®^ is one of the most commonly used biomaterials in breast cancer 3D cell culture [125]. As a recent example, Badea et al. formed multiple MDA-MB-231 breast cancer adenocarcinoma MCTS in a single well by including Matrigel^®^ in the culture media used in LOT and observed a more uniform morphology and greater circularity compared with MCTS formed without Matrigel^®^ [31]. In another study, Gangadhara et al. cultured the ER+/Her2+ breast cancer cell lines BT474 and MDA-MB-361 in Matrigel^®^ and observed compact (BT474) and grape-like (MDA-MB-361) MCTS morphologies, respectively [65]. Their results reaffirmed that 3D culture decreased chemosensitivity to tamoxifen, fulvestrant, and trastuzumab; further, they found that 3D culture promotes switching from the phosphatidylinositide-3-kinase/protein kinase B (PI3K/AKT) to the mitogenic activation of protein kinase (MAPK) signaling pathway, illustrating the importance of 3D culture in studies of cancer biology. Finally, when Matrigel^®^ was compared to lower-cost and easier-to-handle bioinks such as gelatin-alginate and collagen-alginate to bioprint cells to model breast cancer, only the Matrigel^®^ ink (2% v/v) successfully induced MCF10A, MCF10A-NeuN, MDA-MB-231 and MCF7 epithelial breast cancer MCTS formation, indicating a need for alternative bioprinting techniques to decrease cost and increase ease of forming breast cancer MCTS [77].

Matrigel^®^ utility in 3D culture is not limited to breast cancer; it has been shown to induce MCTS formation across many cancer types, including hepatocellular carcinoma [126] and colorectal adenocarcinoma [82]. In the former study, a single-cell suspension of HepG2 cells was seeded in Matrigel^®^ layers and the resulting MCTS were dissociated for further single-cell seeding and formation of MCTS-like cell clusters to isolate cancer stem cells (CSCs) [126]. In the latter, SW1463 and LS174T colorectal adenocarcinoma cells were seeded into Matrigel^®^ for optimization of lipofection-based siRNA delivery to MCTS and organoids [82]. In some cases, even Matrigel^®^-coated wells can induce MCTS formation, as seen recently in a co-culture of tumor-associated macrophages isolated from ovarian cancer patient MCTS and SKOV-3 cells [127].

Matrigel^®^ has been one of the most widely used biomaterials in 3D culture because it is sourced from a tumor ECM and thus closely replicates the in vivo tumor environment, including naturally present growth factors [121]. In fact, its natural source and physiological relevance have made it the most common biomaterial used in cultures of organoids—a more complex and physiologic disease model with different cell types generated from pluripotent stem cells (PSCs) or organ-specific adult stem cells [128]. Matrigel^®^ also serves as an excellent biomaterial in composite approaches because it can naturally add cell adhesion and signaling to scaffolds that otherwise do not contain native ligands [70]. However, differences in the size of the source mouse tumor and tissue preparation techniques can cause intra- and inter-batch variation, limiting its use in drug discovery and studies of cancer biology [129]. The content of the growth factor cocktail in Matrigel^®^ can also cause confounding signaling cascades that make it difficult to use in mechanistic studies, and the ratios of collagen I and HA in Matrigel^®^ do not match those of in vivo human tumors [130]. Furthermore, due to its quick gelation time, working with Matrigel^®^ requires careful handling; refrigerator temperature is often not low enough to prevent gelation, so thawing on ice in the refrigerator and maintaining on ice while handling is good practice [119]. And finally, adjusting the mechanical properties of Matrigel^®^ requires careful consideration, although composite approaches with other natural [131] or synthetic [70] polymers can be further explored to increase tunability.

#### 3.1.4. Alginate/Chitosan-Based Matrices

There are several natural polymers in addition to collagen, Matrigel^®^ and hyaluronic acid that are commonly used to form MCTS. Alginate is a polysaccharide isolated from brown algae and lacks the native ligands required to interact with mammalian cells [67]. Divalent ions such as Ca^2+^ can cooperatively bind adjacent alginate chains, creating interchain bridges that allow gelation of alginate solution into a scaffold for cell culture [132].

Depending on the gel stiffness, alginate hydrogels can induce MCTS formation without modification [36]. In one study, breast cancer MCF-7 cells were cultured in alginate gels with varying stiffness to determine optimal mechanical conditions for MCTS formation. They observed the clearest MCTS formation in gels of 150–200 kPa stiffness compared to 300–350 kPa and 900–1800 kPa despite similar cell viability [36]. However, alginate is commonly functionalized with peptide domains like RGD to encourage cell adhesion [67] or used in a composite approach for 3D culture. For example, chitosan, a biocompatible and biodegradable polysaccharide derived from chitin in crustacean shells [133], is commonly used with alginate to form scaffolds for 3D cell culture. In a recent study, alginate-chitosan scaffolds were used to study the effect of ECM stiffness on prostate cancer response. PC-3, C4-2B and 22Rv1 prostate cancer cells were seeded on the composite scaffolds and MCTS formation was observed in C4-2B and 22Rv1 cells, whereas PC-3 cells only formed grape-like clusters [79].

Because alginate is not bioactive, it is useful in isolating the scaffold’s mechanical contributions to cell behavior, especially because the mechanical properties can be precisely tuned via calcium ion mediated crosslinking [36]. However, alginate is also not sensitive to cell-secreted proteolytic enzymes (non-biodegradable) [134], limiting its use in studies of invasion and migration unless used in a composite approach.

### 3.2. Synthetic Matrices

Synthetic biomaterials have also been used to form tumor MCTS. Synthetic polymers offer several advantages over some natural ECM, including more tunable stiffness and cell ligand density and other biochemical properties. However, these synthetic materials are biologically inactive, and thus must be functionalized with cell adhesion peptide domains (e.g., RGD) to encourage cell adhesion and crosslinked to form biodegradable bonds for cell remodeling of the ECM.

#### 3.2.1. Polyethylene Glycol (PEG)-Based Matrices

PEG is a synthetic polymer widely used in MCTS formation. PEG hydrogels are typically either covalently or chemically cross-linked to form stable, tunable hydrogels. Covalently crosslinked PEG-based scaffolds can be synthesized via chain-growth, step growth, or a mixed mode, and the crosslinking method can affect the number of structural defects and subsequent mechanical properties [135].

PEG hydrogels have been commonly used to form hepatocellular carcinoma MCTS because liver-derived cells must be cultured in MCTS to maintain liver-specific functions [136], and PEG-based hydrogels maintain a high level of this functionality [137]. For example, a hydrogel was formed from 8-arm PEG-SH crosslinked to form disulfide bonds that were sensitive to reduction by the non-toxic reducing agent cysteine. When HepG2 cells were cultured in this gel at varying culture conditions, MCTS formation was observed; furthermore, reduction by cysteine allowed recovery of MCTS for future study [137]. A similar approach was taken to culture HepG2 MCTS in a thermoresponsive hydrogel poly (N-isopropylacrylamide-co-acrylic acid) (PNIPAM) microgel [138]. PEG-based hydrogels have also been used to form MCTS in various other cancer types, including breast cancer [37,70] and lung adenocarcinoma [85].

PEG is an extremely versatile polymer: it is resistant to protein adsorption, biocompatible, can be modified easily via chain length, cell adhesion ligands, and degradable crosslinkers, and is hydrophilic [139], which encourages cell-cell adhesion. Because PEG is synthetic, PEG-based hydrogels benefit from easy tunability, low cost compared to many naturally sourced biomaterials, and reproducibility [140]. However, the lack of biological moieties requires additional functionalization with RGD and protease-sensitive peptides in order to mimic the cell adhesion and degradation by proteases inherently achieved by natural matrices [141].

#### 3.2.2. Peptide-Based Matrices

Peptide nanofibers are formed from self-assembly of peptides that are produced to have specific intermolecular interactions, which can be modified by changing amino acid residues. Hydrophilicity, hydrophobicity, stiffness and rigidity can be modified by using amino acids residues with appropriate characteristics [142]. From a materials science perspective, such tunability at the nanoscale provides a variety of opportunities [143,144,145,146,147,148]. For example, by rational design of these peptides and their side chains, it is possible to prepare fine-tuned one-dimensional nanostructure templates for inorganic material accumulation, which can be used for different applications in medicine and nanotechnology [143,144,145,146]. Moreover, in tissue engineering scaffold formation, modification of peptide sequences allows encapsulation of growth factors or therapeutics for enhanced functionality [147,148]. Peptide nanofibers can form gels (including hydrogels) in aqueous solutions when used at high concentrations (ranging from 4 mM to 12 mM), and there have been multiple studies on encapsulating cells into these hydrogels given their biocompatible and porous structures [149,150,151,152]. 3D structure formation allows mimicking in vivo conditions in terms of cell proliferation, migration, differentiation [153]. Choice of amino acids for construction should be done carefully, otherwise peptide-based surface can be antifouling [154,155]. Amino acids can be arranged to mimic certain epitopes from ECM proteins. Moreover, peptide scaffolds can be modulated for their stiffness which eventually affects MCTS formation [156].

RADA16-I (Ac-RADARADARADARADA-Am) is a peptide sequence used to build 3D scaffolds. Encapsulation of adult progenitor cells into RADA16-I scaffold resulted in the formation of functional hepatocyte MCTS formation [157]. Usage of more hydrophobic peptides KLD12 (AcN- KLDLKLDLKLDL-CONH2) and KFE8 (AcN-KF-EFKFEF-CONH2) caused a more stiff environment which resulted in MCTS formation to a much lesser extend compared to RADA16-I [157]. A self-assembling peptide called bQ13 (Ac-QQKFQFQFEQEQQ-Am) has been shown to promote formation prostate cancer MCTS formation and found to be a better alternative to Matrigel^®^, RADA16-I and Q11 (Ac-QQKFQFQFEQQ-Am) peptides in terms of inducing minimal cytotoxicity to cells [32]. RADA16-I and Q11 peptides also induce MCTS formation, but remain liquid only at acidic pH which causes cytotoxicity during cell encapsulation [158].

Another biomaterial used for 3D culture is the ECM protein dimer called fibronectin (FN), which has been shown to induce MCTS formation in fibroblasts [159] and cancer cells [160]. Furthermore, depletion of FN compromised MCTS formation. RGD is a commonly used peptide motif from fibronectin, collagen, laminin and vitronectin [161,162] that interacts with cell surface integrins [163]. Therefore, RGD has been extensively used to study cell to cell and cell to ECM interactions. A cyclic version of RGD; cyclo-RGDfK and its modification with triphenylphosphonium (TPP) cation was found to facilitate the formation of MCTS in both cancer and healthy cells [164]. Covalent binding of TPP to cyclo-RGD lowered the required peptide amount for MCTS formation given that it provided electrostatic interactions that facilitated improved cell to cell interaction. Interestingly, linear RGD did not have any effect on MCTS formation. Activity of cyclo-RGDfK is thought to be through its interaction with avb3 integrins which results in cell detachment and later MCTS formation [164,165].

In general, peptide-based scaffolds allow easy modifications of their material properties via amino acid substitution and addition [140], addition of epitopes [140], and ability to achieve multifunctionality from a single material [166]. However, each peptide yields its own set of advantages and disadvantages. For example, unmodified RADA16-I has a low mechanical strength, which has been modified through addition of different peptide side chains with limited success [167]. Some hydrogels use insoluble peptides, such as Fmoc-FF, and thus require using solvents like dimethyl sulfoxide (DMSO) during their gelation procedure, which can be toxic to cells in high quantities [168], creating the need for more biocompatible methods of peptide gelation [169].

## 4. Conclusions

In conclusion, this review is intended to provide a practical guide for choosing the right methodology to engineer MCTS based on the advantages, disadvantages, and recent applications of both matrix-free and matrix-dependent techniques. MCTS are a valuable tool in studies of drug screening and cancer biology and fill an important gap between results obtained via 2D culture and animal models and those seen in human clinical trials. However, there are important differences in MCTS use depending on the formation method used. Matrix-free techniques, such as liquid overlay, hanging drop, spinner flask and magnetic levitation often show greater reproducibility and accessibility of MCTS for quantitative end-point measurements. However, these techniques fail to incorporate the cell–ECM interactions which are essential to cell behavior in vivo. Matrix-dependent techniques using natural biopolymers such as hyaluronic acid, collagen, Matrigel^®^, alginate and chitosan, synthetic polymers such as PEG, or peptide-based matrices can often recapitulate these interactions and even induce MCTS formation in cell lines that otherwise will not aggregate. Importantly, when deciding between a matrix-free and a matrix-dependent environment for MCTS formation, researchers must consider existing literature on the accuracy of each formation technique to mimic in vivo cell behavior for the cell type of interest (for example, heterotypic MCTS formed with pancreatic cancer cells in a matrix-free environment show remarkable similarity to in vivo drug resistance) [170]. We provided the cell lines used in matrix-dependent techniques in Table 2. Also, careful consideration of the quantitative analysis that will be ultimately performed on the MCTS is necessary before choosing the technique; for example, an invasion assay will require inclusion of a matrix, but measures of protein expression (e.g., Western blot) or cell population characterization (e.g., flow cytometry) may be simpler without a matrix. Once a suitable technique is chosen, MCTS can serve as a powerful tool for preclinical research ranging from cancer biology to drug discovery and delivery.

In addition to method selection, assessment of MCTS formation requires careful consideration. The precise point of MCTS formation is poorly determined and is a complicated phenomenon due to reasons such as the heterogeneity of the original tumor. A general assessment involves monitoring the cell culture over a period of time and quantifying compactness through image analysis. Additionally, validation of results obtained from MCTS with patient-derived samples can improve MCTS assessments. Moreover, assessments of the identified mechanical properties of MCTS (e.g., stiffness) and correlation with patient-derived samples would provide an important parameter to validate estimates of MCTS formation. The mechanical properties of human tissues would be an excellent target not only to mimic those tissues, but also to study MCTS metastasis in similar tissues. For example, to the best of the authors’ knowledge no studies of the stiffness of the human peritoneum have been reported; these data are essential to create a physiologically relevant matrix for the metastasis of ovarian cancer MCTS in the peritoneal cavity [171]. Identification of MCTS formation parameters that are particularly correlated with patient-derived samples should be considered a crucial aim when targeting a particular application. Considering the complex heterogeneity of real-world tumors, studies on MCTS are still only approximations of the real samples. More studies based on enhancing MCTS heterogeneity better to mimic real samples (via e.g., natural or synthetic polymers) will represent an important advance for cancer research. Overall, MCTS are becoming increasingly adopted as a high throughput, physiologically relevant model for cancer researchers who share the ultimate goal of improving translation of preclinical discoveries to clinical success.

## Figures and Tables

**Figure 1 polymers-12-02506-f001:**
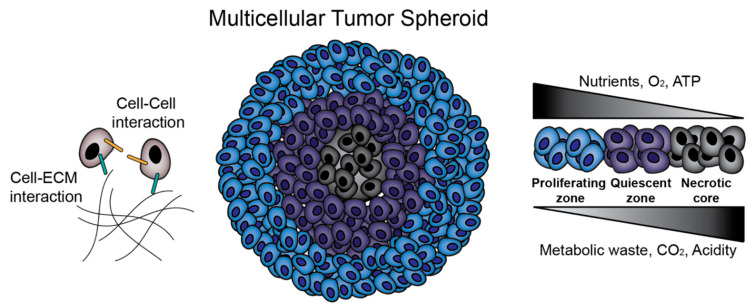
Multicellular tumor spheroids (MCTS) biology. MCTS provide an in vitro platform for the investigation of cell–cell and cell–extracellular matrix (ECM) interactions. Additionally, MCTS mimic in vivo solid tumors in terms of nutrient, oxygen and pH gradients and zone formation.

**Figure 2 polymers-12-02506-f002:**
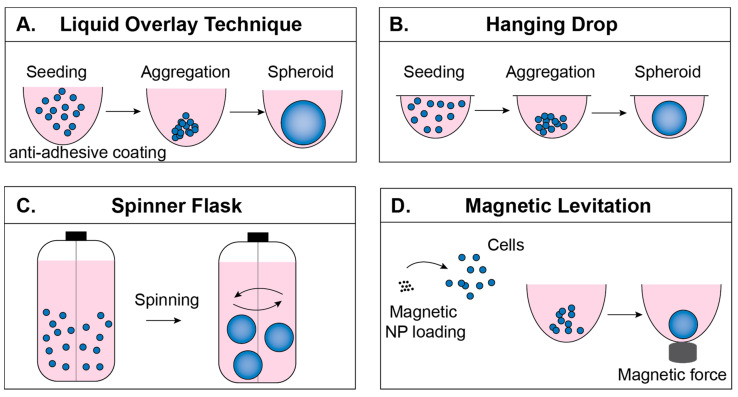
Matrix-free formation techniques. (**A**) In the liquid overlay technique (LOT), cells are seeded onto a surface that prevents adhesion, encouraging cell–cell adhesion. (**B**) In the hanging drop technique, cells are suspended in drops from the underside of a culture plate lid. (**C**) In the spinner flask technique, rotational motion encourages cell–cell adhesion. (**D**) In the magnetic levitation technique, cells take in magnetic nanoparticles and are aggregated by magnetic force.

**Table 1 polymers-12-02506-t001:** Cost, formation timeline and uniformity of matrix-independent and matrix-dependent techniques.

Technique	Cost	Days to Form	Diameter + Std. Dev
Liquid Overlay Technique (LOT)	$62/24-well plate (Aggrewell™, StemCell™ Technologies, Vancouver, BC, Canada)	24 h (OVCAR8) [12] 24 h (RT4) [27]	274.08 ± 13.98 μm to 492.14 ± 25.32 μm [27]
Hanging Drop	_____	24 h (RT4) [27] 12–24 h (MCF-7) [28]	340.92 ± 16.98 to 563.97 ± 28.53 μm (RT4) [27] 205 ± 20 µm (MCF-7) [28]
Collagen	$222/100 mg (Rat tail collagen type I, Sigma Aldrich, St. Louis, MO, USA)	Day 3 (A2780) [29]	211.75 μm ± 16 μm (A2780) [29]
Hyaluronic Acid	$175/g (<10 kDa to >1.8 MDa Sodium Hyaluronate, LifeCore Biomedical, Chaska, MN, USA)	Day 4 (LNCaP) [30]	_____
Matrigel^®^	$314.62/10 mL (Corning^®^, Corning, NY, USA)	Day 3 (MDA-MB-231) [31] Day 7 (LNCaP) [32]	120.2 μm ± 3.8 μm (LNCaP) [32]
Alginate	$127/kg (Sodium Alginate, Sigma Aldrich)	Day 7 (U-251) [33]	99 ± 18.9 μm (MCF-7) [34]
Chitosan	$68.60/50 g (medium MW, Sigma Aldrich)	Day 3 (U87 and U118) [35] Day 7 (MCF-7) [36]	_____
Poly(ethylene glycol) (PEG)	$50–250/kg (Sigma Aldrich)	Day 3 (MCF-7) [37] Day 7 (LNCaP) [38] (OV-MZ6, SKOV3) [39]	_____
Peptide-based	$333 (PuraMatrix™ RADA16-I, Corning^®^)	Day 7 (LNCaP, RADA16-I, bQ13, Q11) [32] Day 5 (MDA-MB-435S) [40]	112.2 ± 5.0 μm in bQ13 (LNCaP) [32] 114.6 ± 5.5 μm in RADA16-I (LNCaP) [32] 111.6 ± 4.7 μm in Q11 (LNCaP) [32]

**Table 2 polymers-12-02506-t002:** MCTS studies using matrix-dependent approaches by cancer type.

	Matrix	Collagen	HA	Matrigel^®^	Alginate/Chitosan	PEG	Peptide-Based
Cancer							
Glioblastoma multiforme (GBM)		U87 (bovine, isolated from spongy bone) [72]; primary (bovine, isolated from spongy bone) [72]	U87 (5 wt%, 60 kDa) [26]; D456 (5 wt%, 60 kDa) [26]	_____	U118 (1.3% w/v chitosan-PEG) [35]; U87 (1.3% w/v chitosan-PEG) [35]; U251 (2% w/v) [33]	U87 [73]	_____
Ovarian Cancer		A2780 (0.125% agarose, 10% rat tail Type I collagen, 1% alginate) [29]	_____	SKOV-3 (3% v/v, growth factor reduced) [74]	A2780 (0.125% agarose, 10% rat tail Type I collagen, 1% alginate) [29]	OV-MZ-6 (2% w/v) [39]; SKOV-3 (2% w/v) [39]	A2780 (0.5% w/v RADA16-I) [75]
Breast Cancer		MCF-7 (2% w/v sodium alginate, 1.5 mg/mL rat tail collagen type I) [34]	T47D (1.1% HA, 289 kDa) [68]; MDA-MB-231 (>106 kDa) [76]	MDA-MB-231 (2.5%) [31], (2%) [77]; BT474 (10% v/v) [65]; MDA-MB-361 (10% v/v) [65]; MCF10A (2% v/v) [77]; MCF10A-NeuN (2% v/v) [77]; MCF-7 (2% v/v) [77]	MCF-7 (2% w/v sodium alginate, 1.5 mg/mL rat tail collagen type I) [34]; MCF-7 (0.5%-2% w/v) [36]	MCF-7 [37]	MDA-MB-453S (1% w/v RADA16) [40]
Prostate Cancer		LNCaP [78] (porcine type A gelatin functionalized with methacryloyl)	LNCaP (20 mg/mL, 500 kDa) [30], PC-3 (5–10% HA-MA) [24]	LNCaP [32]	C4-2B (2, 4, 6 wt% chitosan, alginate) [79]; 22Rv1 (2, 4, 6 wt% chitosan, alginate) [79]	LNCaP (1.5% w/v) [38]; PC-3 [37]	LNCaP (1.45 mM, 5 mM and 15 mM of RADA16-I, bQ13, Q11) [32]
Colorectal Cancer		HT-29 (1.8 mg/mL, rat tail Type I collagen) [80]	_____	LOVO COLO-205 CACO-2, COLO-206F, DLD-1, HT-29, SW-480 [81]; LS174T [82]; SW1463 [82]	HCT116 (2% w/v alginate) [83]	HT29 [37]	_____
Lung Cancer		A459 (2 mg/mL) [84]	_____	344SQ (2%, growth factor reduced) [85]	_____	344SQ (10% w/v PEG-PQ, 3.5 mM RGDS) [85]	_____

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
