# Peer review of "Natural and Synthetic Biomaterials for Engineering Multicellular Tumor Spheroids"

_polymers, 2020, doi:10.3390/polym12112506_

Round 1
Reviewer 1 Report
This review paper purposed that the current state of biomaterial-based multicellular tumor spheroids formation and their applications to the cancer field, drug screening and solve the major problem between preclinical trials using 2D cell culture, mouse models, and clinical trial research. Moreover, both matrix-free and matrix-dependent techniques were determined as a practical guide to consider for choosing the better approach for the research targets, including the advantages and disadvantages of each method. However, several similar reviews have been reported in the last few years. The authors should address the below comments before this paper can be considered for acceptance.
- As the title of the paper stated, the “Natural and synthetic biomaterials for engineering multicellular tumor spheroids.” However, only biomaterials related to matrix-dependent methods were reported and summarized, which is not true, as there are several biomaterials being used on matrix-independent methods. For instance, in the section of liquid overlay techniques, only the ULA platform was highlighted, yet, several microwell-based methods required different levels of biomaterials used to prevent cell attachment to form spheroids. None of these aspects was covered. I suggest the relevant works, with the focus on the adhesion repellent being used, should be cited. Below two papers are a good direction for the authors to cite and expand some related microwell works.
[1] Tu TY, Wang Z, Bai J, Sun W, Peng WK, Huang RYJ, et al. Rapid prototyping of concave microwells for the formation of 3D multicellular cancer aggregates for drug screening. Adv Healthc Mater 2014;3:609–16. https://doi.org/10.1002/adhm.201300151.
[2] Chiu CY, Chen YC, Wu KW, Hsu WC, Lin HP, Chang HC, et al. Simple in-house fabrication of microwells for generating uniform hepatic multicellular cancer aggregates and discovering novel therapeutics. Materials (Basel) 2019;12:3308. https://doi.org/10.3390/ma12203308.
- While the authors have indicated different techniques utilized for the engineering of MCTS, I suggest a few parameters that should be included, i.e. the cost, days required to form MCTS, uniformity, etc. These aspects will be interesting for the readers.
- Only a part of the references was cited in Table 1. For example, the reference in lines 215 [83], 291 [101], line 397 [125] were all not included in Table 1. The authors should check the missing citations.
- As different sources of type I collagen can pose different effects on the cultured cells, the authors should indicate the species origin of the collagen used in both of Table 1 and the statement in line 218 “Type I collagen can convert cancer cells into cancer stem cells…” In addition, as the concentration of the collagen would also affect the growth conditions of the cells, the authors should consider putting in this information.
- In 411, the authors stated that the peptide nanofibers can form gels at high concentrations. The authors should indicate the concentration ranges being applied from the studies. In fact, the authors should consider include the information of the polymer/hydrogel concentration used in the cited references, especially for the matrix-dependent approaches in Table 1.
- Also, the limitation, applications, and future perspectives are all important components that should be included to strengthen the current version.
Reviewer 2 Report
This review aims at evaluation of biomaterial-based MCTS formation, which is a translational tool in biomedical research. The purpose of the manuscript is to provide a practical guide for choosing a technique for MCTS formation in vitro. The authors describe advantages and disadvantages of the currently used biomaterial applications, with a focus on the past five years of cancer research.
The review is a thorough and detailed summary of the current methods utilized to prepare in vitro MCTS with a practical view. The introduction is the strength of the MS. The figures are expressive and ease the reading of the manuscript, that is often hard to follow due to wording rather used in spoken english. It would be beneficial to add information about high troughput tools that are already available for matrix-free MCTS (e.g. hanging drop). The natural biopolymer matrices section is a thorough and well presented dicussion of biomatrix applications and their limitations with appropriate references.
Specific comments
line97 A brief protocol would be helpful for the reader.
line135 How does it ease tracking and imaging?
line 159 Invasion and angiogenesis studies need a matrix, but can be achieved by subsequent embedding, not a preprequisite for using matrix-dependent spheroid formation.
line126 Please add examples and references
line302 A short description of basement membrane would be beneficial here
line410 Please list examples for „variety of opportunities”.
Round 2
Reviewer 1 Report
Major: The below paragraphs are repeated in parts 2.2 and 2.3.
- Line 149-159 and 161-171, “The spinner flask technique allows cell to cell interaction by preventing the settling of the cells through continuous spinning [54], and leads to MCTS formation (Figure 2C). For example, HepG2 hepatocellular carcinoma cell suspensions were seeded at 1 x 105 cells/mL into 500 mL siliconized spinner flasks and stirred at 60 rpm for 4 to 6 weeks. The resulting MCTS were 200-300μm diameter and were transferred into agarose-coated well plates for further processing [55]. In another study, rat liver hepatocytes were cultured at 5 × 105 cells/ml in a 250mL spinner flask for 6 hours, resulting in 80% aggregate formation [56]. The speed of the continuously rotating flask is important to consider: low speeds might result in settling of the cells and high speeds might harm the cells due to shear stress. One advantage of using the spinner flask is the ability to produce large quantities of MCTS. However, during continuous spinning MCTS formation cannot be visualized [54], which is important in monitoring morphology and size.”
Minor: Be consistent with the units and spacing.
- Line 162-166, “For example, HepG2 hepatocellular carcinoma cell suspensions were seeded at 1 x 105 cells/mL into 500 mL siliconized spinner flasks and stirred at 60 rpm for 4 to 6 weeks. The resulting MCTS were 200-300μm diameter and were transferred into agarose-coated well plates for further processing [55]. In another study, rat liver hepatocytes were cultured at 5 × 105 cells/ml in a 250mL spinner flask for 6 hours, resulting in 80% aggregate formation [56].”
- Line 268, “Rat tail type I collagen-coated on well-plates at 40-60 μg/mLhas been shown to convert cancer cells into cancer stem cells [97], …”
